# Postmortem Bacteriology in Forensic Autopsies—A Single Center Retrospective Study in Romania

**DOI:** 10.3390/diagnostics12082024

**Published:** 2022-08-21

**Authors:** Iuliana Diac, Arthur-Atilla Keresztesi, Anda-Mihaela Cerghizan, Mihai Negrea, Cătălin Dogăroiu

**Affiliations:** 1Mina Minovici National Institute of Legal Medicine, 042122 Bucharest, Romania; 2“Fogolyan Kristof” Emergency County Hospital Sfantu Gheorghe, Covasna County Institution of Forensic Medicine, 520045 Covasna, Romania; 3Medical Clinic, George Emil Palade University of Medicine, Pharmacy, Science and Technology of Targu Mures, 540043 Mures, Romania; 4Department of Public Health, Faculty of Political, Administrative and Communication Science, “Babeș Bolyai” University, 400084 Cluj Napoca, Romania; 5Department of Legal Medicine and Bioethics, Carol Davila University of Medicine and Pharmacy, 050474 Bucharest, Romania

**Keywords:** postmortem bacteriology, medico-legal autopsy, sampling patterns

## Abstract

Postmortem bacteriology examinations have been a controversial topic over the years, though the value of postmortem bacteriology cultures remains promising. The aim of this study was to review the postmortem bacteriological sampling procedures and results in a single centre in Bucharest over a period of 10 years. Material and methods: The present study was a retrospective, single-center study, performed at the Mina Minovici National Institute of Legal Medicine in Bucharest, Romania, from 2011–2020. Results: Postmortem bacteriology was requested 630 forensic autopsies, 245 female (38.9%) and 385 male (61.1%), age range 0 and 94 years, median age of 52 years. Deaths occurred in hospital for 594 cases (94.3%) and out-of-hospital for 36 cases (5.7%—field case). Blood cultures were requested in the majority of cases, followed by tracheal swabs and lung tissue. In-hospital and out of hospital deaths did not differ significantly regarding the number of microorganisms identified in a positive blood culture. Postmortem bacteriology cultures of the respiratory tract showed a statistically significant association to microscopically confirmed lung infections. Conclusions Postmortem sampling for bacteriology testing in our center in Bucharest is heterogeneous with a high variation of patterns. A positive blood culture result for *Staphylococcus* species without the identification of a specific microorganism is more likely due to postmortem contamination.

## 1. Introduction

Postmortem bacteriology (PMB) examinations have been a controversial topic over the last century, with studies demonstrating divergence from ante-mortem results [1] or a poor correlation between the type of infection and/or indicators of infection and the microorganisms recovered at autopsy [2], due to events occurring after death. Of concern is a false-positive result for a potentially pathogenic microorganism without being the expression of true antemortem bacteraemia [3]. 

Aiming to assess the value of postmortem microbiology cultures based on existing literature, Riedel [4] discusses the two main theories explaining bacterial growth in postmortem blood and tissue cultures: agonal spread and postmortem bacterial transmigration, respectively. Additionally, he acknowledges the possibility of iatrogenic contamination of postmortem cultures during the procurement of samples. Based on the reviewed literature, he concludes that postmortem cultures recovered from autopsies performed in the first 48 h after death have a low probability of agonal spread and of postmortem bacterial transmigration. Agonal spread is the process in which bacterial invasion occurs during the process of dying or during resuscitation attempts due to ischemic damage induced on mucosal surfaces. Postmortem translocation is the bacterial migration from the mucosal surface into the blood and internal organs after death [5]. Inappropriate technical measures such as the delayed cooling of the corpse or sampling procedure, using non-sterile tools or operating in non-sterile environments, may cause contamination [6]. Monomicrobial cultures of a typical opportunistic and/or pathogenic microorganism obtained from blood or tissue samples can be considered a true indicator of infection, in contrast to polymicrobial growth and/or the presence of typical contaminant organisms that are likely a result of iatrogenic contamination during biospecimen procurement or postmortem bacterial transmigration [4].

Contamination during sampling procedures and postmortem translocation, endogenous bacterial multiplication and migration into blood and tissues after death, were investigated systematically in one hundred medico-legal autopsies of out-of-hospital deaths. A prospective, single-centre study performed at the University Centre of Legal Medicine in Lausanne, Switzerland, during 2008–2014 included a complex protocol based on collecting radiology, histology, bacteriology, and biochemistry results [3]. This study showed that contamination during the sampling procedures was characterised by the presence of a single or multiple microorganisms, frequently *Staphylococcus* epidermidis or another coagulase negative *Staphylococci,* alone or in association with commensals of the skin/mucosal membranes/mouth/upper respiratory tract. Moreover, bacterial translocation was suspected if one or multiple microorganisms originating from the gastrointestinal tract—such as Gram-negative members of the *Enterobacteriaceae* family (*Escherichia*, *Enterobacter* and *Klebsiella*) were present, and/or bacterial translocation gram-positive commensal(s) of the human gastrointestinal tract (*Enterococcus* groups or *Lactobacilli)* [3]. Other research most commonly revealed coagulase negative staphylococci, as a contaminant, sometimes along with *Micrococcus*, *Corynebacterium*, or *Propionibacterium*, and similar translocation microorganisms [7].

The purpose of the study is to review the postmortem bacteriological sampling procedures and results at our forensic institute over a period of 10 years.

## 2. Materials and Methods

We performed a retrospective, observational, single-centre study, at the Mina Minovici National Institute Legal Medicine in Bucharest, Romania, based on postmortem bacteriology results and corresponding autopsy records collected from 2011 to 2020. We retrieved all the cases in which bacteriology examination were solicited during the medico-legal autopsy. The autopsies were requested according to the criminal law—violent or suspected to be violent death. Postmortem bacteriological analysis was performed as part of the medico-legal investigations as an additional examination not part of the standard autopsy protocol in our centre, based on the clinical judgment and in accordance with the findings in each case, on a case-by-case-basis. The cases included in this study originated from forensic practice, both in-hospital and out-of-hospital (field). Data were collected from the request for analysis, the bacteriological and autopsy reports as well as from medical records, as showed in Table 1.

For PMB sampling in our centre, there is no standardised protocol; routinely, in our centre, the procedure for collecting is specimen oriented as requested by the forensic pathologist, and the bio samples are obtained by a procedure-trained autopsy assistant. Blood samples are generally collected by aspiration of blood from the heart chambers after opening the pericardial sac using a sterile needle and syringe; exceptionally, the blood is collected from a peripheral source such as the femoral vein. Swabs, tissue samples and fluid collection samples are retrieved so as to minimise contamination. All samples are transferred to the laboratory on the day of collection or the following day at the latest. No specimens were excluded due to insufficient sample volume.

Blood and tissue cultures were analysed based on current standards of accepted procedures by the same laboratory—Victor Babes Private Medical Clinic—in Bucharest, an external collaborator. 

Collected data was included in an electronic data base using Microsoft Excel 2010 software. Statistical analysis was performed using IBM^®^ SPSS Statistics for Windows version 24.0. Continuous variables were described by mean and standard deviation or by median and range; categorical data were expressed as frequencies. Descriptive data were compared using the Chi-Square test, while comparison of differences between groups was performed by using ANOVA. A *p*-value below 0.05 was considered statistically significant.

### Ethics Statement

Relevant ethical issues were identified and discussed with the institutional ethical committee. During the 10-year period, all the bacteriological tests performed were deemed necessary in order to establish the cause of death. They were requested by the medico-legal specialist to whom the case was assigned. All medico-legal autopsies in Romania are mandated by police or state prosecutor. Institutional access for the archives was obtained. All data regarding the results of the blood and postmortem samples was anonymised prior to analysis. The informed consent of the relatives for data was not deemed feasible considering the time frame from which the data was retrieved.

## 3. Results

For the ten-year period (2011–2020), we found a number of 630 cases in which bacteriological investigations were solicited. Bacteriological analysis was requested in few cases compared to the total number autopsies—between 1.89% and 5.59% for each year. The demographic characteristics of the study population were as follows: 245 female (38.9%) and 385 male (61.1%), age range 0 and 94 years, with a median age of 52 years. We divided the study population into five age groups (0–19 years, 20–39 years, 40–59 years, 60–79 years, 80–100 years) and assessed the mean days of the hospital stay and ICU stay for each group; thus, for our study population there was no significant difference between groups for the hospital stay (*p* = 0.225) or ICU stay (*p* = 0.764). Regardless of age group, our study cases were predominantly hospital deaths; nonviolent and infection was listed in the cause of death in around 80% (from 78.3% for 20–39 years to 87.1% for 80–100 years).

Deaths occurred in the hospital for 594 cases (94.3%) and out-of-hospital for 36 cases (5.7%—field case). For sample type, bacteriological and pattern analysis, we used all 630 cases, but for autopsy findings and hospital records we were unable to locate 84 autopsy records, thus including in the statistical assessment only 546 cases (522 hospital cases, 24 field cases). This was partially due to the fact that some autopsy records were requested and in use in ongoing civil or criminal lawsuits, or not yet completed for the year 2020 at the time of data collection. The manner of death was omitted in 16 (2.9%) cases, listed as nonviolent in 341 (62.5%) cases and violent in 189 (34.6%) cases. Regardless of the manner of death, infection was mentioned as a cause of death in over 80.0% of cases (81.9% nonviolent, 85.7% violent), as showed in Table 2.

Out of the 630 medico-legal autopsies included in the study, a total of 1166 samples were collected (Table 3). The minimum number of samples per case was one and the maximum was six with a median of two samples/case.

The sampling patterns varied considerably: over the ten-year data, we identified 104 different patterns for hospital cases and 19 patterns for field cases. For both subgroups, postmortem blood cultures were the most frequent assay, followed by tracheal swabs/lung tissue samples, and lastly, wound swab/skin tissue sample for hospital cases and collection samples for field cases, as represented in Figure 1. 

For sample type association and cover of the study population, we used a TURF analysis with a maximum variable combination of six (6) biological samples/case. The combination of the lung tissue and tracheal swab, pleural fluid and blood cultures covered 81.0% of study cases, and when up to six biological sample were taken, cultures from lung tissue and tracheal swab, abscess/collection, pleural fluid, blood, faeces, skin tissue and wound swabs, we found that up to 90.3% of cases in our study population were covered—Figure 2.

From the solicitation sheet, we obtained the date of death and the date of autopsy, thus allowing us to compute the postmortem interval for sampling (days). The median value for the postmortem interval is 2 days with a minimum of one day and a maximum of 8 days.

The sampling was performed within the first two days (48 h) in 334 autopsies (53.0%) and within the first 4 days in 579 autopsies (91.9%) of cases, as reflected in Figure 3. Exceptionally, for less than 1% of cases, the autopsy was performed later than one week. The postmortem interval for sampling was significantly shorter for field cases compared to hospital cases (mean postmortem interval: hospital cases 2.70 ± 1.272 days, field cases 1.83 ± 1.108 days, *p* < 0.001).

Out of the 457 requested blood cultures, 41 (9.0%) were negative, sampled between 1 and 6 days postmortem with a median of 2 days. In the other 416 cases, the blood cultures were positive for one to six microorganisms, with the majority positive for one or two microorganisms. Out of the 190 tracheal swab and lung tissue sample cultures requested, 25 (13.1%) were negative. For the positive cultures, one to five microorganisms were identified, with 152 (92.1%) positive for one to three microorganisms. Pleural fluid samples were collected in 70 cases, out of which 16 cases (22.9%) were negative. Positive results were found for one to four microorganisms/case, with a majority for one or two microorganisms in 44 cases (81.5%). Peritoneal fluid samples were retrieved in 75 cases, with 33 (44%) negative and positive cultures revealing one to three microorganisms/case. The faeces samples were collected in 52 cases with 28 (53.8%) negative cultures. Among the positive cultures, 20 (83.3%) identified *Clostridium difficile*. All five heart valve tissue samples were positive for one to four microorganisms. Pericardial fluid cultures were negative in 22 (56.4%) cases and positive results found one to three microorganisms. Cerebrospinal fluid cultures were negative in 25 (37.4%) cases, with the majority of positive cultures (57.14%) identifying one microorganism. Meningeal and brain tissue samples were negative in (29.2%) of cases, with the majority of positive cultures (12–70.5%) identifying one microorganism. Wound swabs and skin tissue cultures were all positive, identifying one to five microorganisms. Urine cultures were negative in 11 (30.5%) of cases, with 92% of positive cultures identifying one or two microorganisms. Synovial fluid cultures were positive in two of the three cases sampled. Fluid collection cultures were negative in the 13 (28.9%) cases in which samples were retrieved, with the majority of positive cultures identifying only one organism—Figure 4. 

During the ten-year period, the family of bacteria retrieved from blood cultures in the majority of cases was *Enterobacteriaceae—*Table 4. As a pathogen from this family, *Klebsiella pneumoniae* was the microorganism most frequently identified during the ten-year period. *Enterobacter aerogenes*, also known now as *Klebsiella aerogenes,* appears in our post-mortem bacteriology results for the first time in 2014, in a relatively high number of cases (32.14% of positive cultures); it appears constantly through the following years up to 2016, after which it dwindles (between 0–3 positive results/year). *Acinetobacter baumannii* and *Pseudomonas aeruginosa* were also in the top three pathogens for several years. *Staphylococcus aureus* or other staphylococci were not commonly identified blood microorganisms in our study cases. 

The number of microorganisms/culture did not vary significantly with case type; thus, 70.8% of blood cultures for hospital cases were polymicrobial, while 31.3% were monomicrobial for field cases (*p* = 0.86). Comparison between microorganisms identified in blood cultures and the postmortem interval identified only one statistically significant association, respectively; the result was the *Staphylococcus* species (spp.—olymicrobial-microorganisms from the family) (median negative results 2.68 ± 1.28 days and positive 3.52 days ±1.209, *p* = 0.003)—Figure 5. 

Dividing our population by case type (field or hospital), we performed a stratification analysis using Cochran’s and Mantel-Haenzel Statistics for the association of postmortem lung or tracheal swab cultures and histologically confirmed lung infection. In the hospital group, microscopically confirmed lung infection coincided with positive cultures for respiratory samples in 128 cases (89.5%), while absence of lung injury was associated with negative culture in 4 cases (66.7%). For the field group, positive cultures and microscopically confirmed lung infection were identified in four cases (80.0%), with no significant difference in culture results for the absence of lung injury. In the hospital group, a positive postmortem lung or tracheal swab culture result was significantly associated with a microscopically confirmed lung infection (*p* = 0.001) in contrast to our field group, where the postmortem organ specific culture results demonstrated no significant association to the confirmed lung injury (*p* = 0.81). Regardless of the case type, the postmortem bacteriology result of respiratory system related samples demonstrated a statistically significant association to the microscopically confirmed lung infection (*p* < 0.001). Without the case type stratification (hospital or field case) of study cases, the overall probability of obtaining a positive bacteriology result for lung and tracheal swabs samples is 12.37 (95% CI: 3.15–48.57) times greater in the microscopically confirmed lung infection. The results of the Mantel-Haenszel Common Odds Ratio Estimate show that after controlling for case type, the probability of a positive culture in a microscopically confirmed lung infection is 10.21 times greater in hospital cases than in field cases (*p* = 0.004, ORMH = 10.21 (95% CI: 2.11–49.26)—Figure 6.

*Enterobacteriaceae* was the family of bacteria retrieved from lung cultures in the majority of cases (66.67%), with similar percentages for *Acinetobacter baumannii* and *Pseudomonas aeruginosa* combined (63.64%).

Wound swab or skin tissue samples were only collected for hospital cases, and macroscopic or microscopic signs of skin lesions were reported in 55 (84.6%) of the cases with positive cultures. There was no association between the postmortem interval and number of microorganisms identified (*p* = 0.122). In 85.5% of cases with autopsy-reported skin lesions, polymicrobial cultures were identified (*p* = 0.004). In positive cultures, *Acinetobacter baumannii* and *Pseudomonas aeruginosa* were most frequently identified (62.0%), followed by *Enterococcaceae* (56.10%), irrespective of the postmortem interval.

The predominant microorganism identified in cultures from pleural, peritoneal and pericardial fluid revealed that they belonged to the *Enterobacteriaceae* family (25.64–58.57%, respectively). For 20 of the cases, both blood and fluid collection cultures were taken and were both positive in 13 (76.5%), though without statistical significance (*p* = 0.133), probably due to the small number. 

In our study population, no bacterial family demonstrated a statistically significant association to the postmortem interval with similar grouped median days for positive and negative cultures (grouped median number of days: negative cultures 1.88–3.75 days and positive culture 1.33–2.72 days).

Some postmortem blood culture results were associated with other sample cultures, as follows: peritoneal fluid (*p* = 0.010), tracheal swabs and lung samples (*p* < 0.001) for *Enterobacteriaceae*; tracheal swabs and lung samples (*p* < 0.001), pleural or peritoneal fluid (*p* = 0.027), and fluid collections cultures (*p* = 0.035) for *Pseudomonas*; tracheal swabs and lung samples (*p* < 0.001) and cerebral spinal fluid (*p* = 0.008) for *Enterococcaceae*; brain/meningeal fluid (*p* = 0.003), fluid collections and pleural fluid (*p* < 0.001) for *Streptococci*; cerebral spinal fluid (*p* = 0.022) and peritoneal fluid (*p* = 0.042) for *Staphylococci*.

We collected data regarding the bacteriology results prior to death from autopsy records for 233 (42.6%) of the in-hospital deaths and 21 cases (9.1%) were negative. In this subgroup, there was an association between positive in-hospital bacteriology results (regardless of sample type) and postmortem bacteriology results for *Clostridium difficile* (*p* < 0.001), *Klebsiella pneumoniae* (*p* = 0.035) and *Proteus mirabilis* (*p* = 0.018). In the cases with negative in-hospital bacteriology cultures, only *Staphilococcus aureus* was identified in postmortem cultures—28.6% (*p* = 0.022).

## 4. Discussion

In Europe, a practice survey gathering information from 16 countries demonstrated that even if there are some common practices and local bacteriology sampling protocols, there is currently no standardised sampling procedure in PMB [8]. For different clinical scenarios, such as the autopsy procedure for Sudden Unexpected Death in Infancy (SUDI) and in Childhood (SUDC), existing literature recommend protocols that standardly include samples from the respiratory and the central nervous systems, blood, spleen, heart, other tissues, and colon [7,9,10,11]. 

Being a retrospective study, we had no control over the autopsy cases for which bacteriological examination was requested by the case legal-medicine specialist. From the records, the vast majority of the cases for which this examination was requested were hospital cases, which were severely wounded/burned patients that received medical care but with an unfavourable outcome. The lack of a PMB sampling protocol is also illustrated by the heterogeneity of sampling patterns identified in our data (104 different patterns for hospital cases and 19 patterns for field cases over a period of 10 years). Moreover, a combination of just six variables (cultures from lung tissue and tracheal swab, abscess/fluid collection, pleural fluid, blood, faeces, skin tissue and wound swabs) covered over 90% of our cases.

Most studies advise that bacteriological sampling should be performed within the first 24 h post-mortem [12,13] or in 48 h [14]; other studies suggest relevance up to 5 days postmortem [15]. Our data revealed a median postmortem interval of 2 days (48 h) before sample collection. In Romanian hospitals, deceased patients remain on the ward for two hours after death; afterwards, the bodies are promptly stored in refrigerated units. Legal jurisprudence requires out-of-hospital deaths that are deemed violent, suspected of being violent or unexplained be stored in appropriate refrigerated units as soon as possible. This difference in procedure may explain why the postmortem interval for sampling was significantly shorter for field cases compared to hospital cases.

We aimed to assess if the results of the bacteriological examinations were considered relevant and included in causes of death (final, intermediate or initial) and the correlation with histological results or if they were considered contamination of any kind and not included in the final conclusions of the autopsy report

Just as with previous reports, our data supports contamination rather than a true pathogen, when the PMB result is *Staphylococcus* spp. (*p* = 0.003), a polymicrobial blood culture from the family [7,8,16]. Recent animal model studies demonstrate that from day 2 up to day 18 postmortem, *Staphylococcus* spp. are predominant genera registered as a contaminant [17]. Other known contaminants or microorganisms known to be recovered in PMB results due to postmortem bacterial translocation [8] were not associated with the postmortem interval or absence of infection on histopathological analysis (*p* > 0.05). 

In accordance with prior studies [18,19], PMB is a tool for the identification of an etiological agent, particularly for pneumonia, as PMB positive samples collected from the respiratory tracts were associated with a histopathologically confirmed lung infection. The probability of a positive culture in the microscopically confirmed lung infection is 10.21 times greater in hospital cases than in field cases; most lung tissue/pleural fluid cultures were positive for *Enterobacteriaceae* including *Klebsiella pneumoniae,* followed by other health-care associated related pathogens such as *Acinetobacter baumannii* and *Pseudomonas aeruginosa*, irrespective of postmortem interval.

Our data demonstrates that hospital or field cases do not differ significantly in the number of microorganisms identified in positive blood cultures. Although in postmortem bacteriology polymicrobial growth is considered contamination [16,20,21], with some studies reporting mixed growth bacteria as genuine pathogens [22], for our study population most wound swab or skin tissue samples were polymicrobial regardless of postmortem interval, probably due to the circumstances of death (burn victims, trauma victims with prolonged hospital stay, immobilised patients with pressure ulcers), known for polymicrobial infections [23,24,25].

Although *Clostridium spp*. are fast-growing members of postmortem microbial ’ contaminants, including *Clostridium difficile* [26], our positive results for the latter were in faeces samples, with correlation to known infection prior to death. Though bacteriological assays prior to death were not available for all cases, our results also demonstrated PMB confirmation of previously known infections with *Enterobacteriaceae* (*Klebsiella pneumonia*, *Proteus mirabilis*), regardless of postmortem interval. 

Previous studies suggested that postmortem spleen cultures may be useful in assessing the efficacy of treatment of antemortem bacteraemia [27]; however, spleen samples were scarcely retrieved in our study cases, thus having insufficient data in assessing the relevance of this sample type. PMB for in-hospital deaths can be used to assess treatment response [28]; nevertheless, the rapid deterioration of the patient possibly had a confounding effect of antimicrobial treatment on the post-mortem cultures results [18]. Our study did not aim to assess treatment response, and the data retrieved from the medical records did not include the in-hospital clinical and laboratory onset of infection or antibiotic treatment received, thus conferring a limitation in correlation to the antemortem results. The postmortem assessment of treatment efficacy was a limitation of our study.

The collection of specimens from at least two different sampling sites was more likely to identify an underlying (potentially fatal) infection, as was the case with blood cultures and lung samples or collection. 

Another limitation was that we did not have at our disposal the case doctor’s reasons to request the examinations; as it is performed on a case-by-case-basis, one can assume it was to confirm/identify an etiological agent or to assess a treatment result. We did not intend to profile immune-compromised patients, this being a topic of extensive research in clinical fields for severity and prognostic scales. Regarding the types of medical units included in our analysis, most cases came from emergency hospitals that admit a large number of patients with life-threatening conditions.

## 5. Conclusions

Postmortem sampling for bacteriology as an additional examination requested on a case-by-case-basis in our centre was heterogeneous, with a high variation of patterns, as we identified 104 different patterns for hospital cases and 19 patterns for field cases.

A *Staphylococcus* spp. positive blood culture without the identification of a specific microorganism was more likely due to postmortem contamination rather than having the role of an etiologic agent, in accordance with previous research on contamination during the sampling procedure.

Correlation between postmortem culture results, taken from different anatomical sampling sites, with autopsy and histological findings, can enable the forensic pathologist to identify the etiologic agent of antemortem infection and offer significant help in establishing the precise cause of death. Our results revealed that infection was listed in the cause of death in 83.2% of assessed cases for which postmortem bacteriology was requested.

Our results were promising in reconfirming a microorganism identified in the hospital setting, with limitations due to difficulty in recovering data; only 42.6% of assessed in-hospital deaths had available antemortem bacteriology results.

## Figures and Tables

**Figure 1 diagnostics-12-02024-f001:**
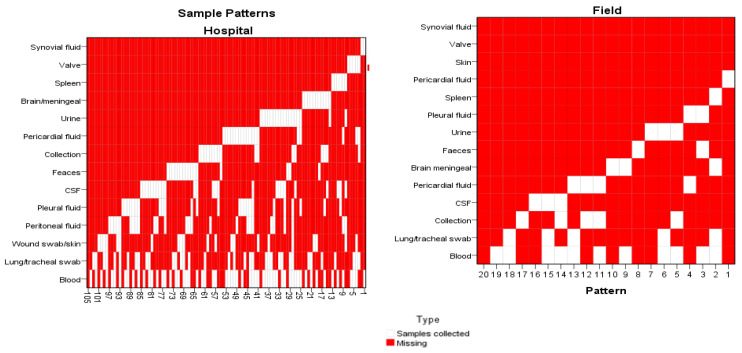
Postmortem sampling patterns. Sampling patterns identified by study group: for culture types we have 14 variables: blood, lung/tracheal swab, wound swab/skin tissue, pleural fluid, peritoneal fluid, pericardial fluid, cerebrospinal fluid (CSF), synovial fluid, collection/abscess, spleen, heart valve, urine and faeces. Missing values analysis identified in the study population 104 different patterns for hospital cases and 19 patterns for field cases.

**Figure 2 diagnostics-12-02024-f002:**
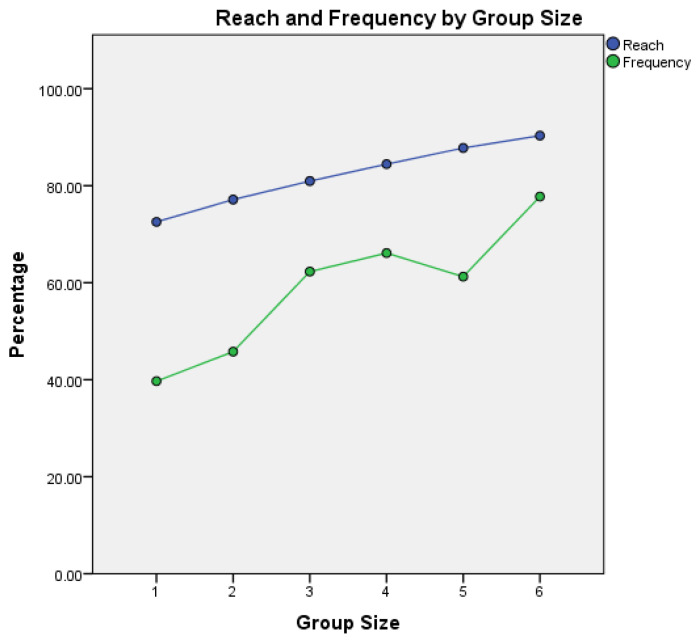
Total Unduplicated Reach and Frequency (TURF) analysis for sample type association in the study population, best reach and frequency by group size, and maximum variable combination of six (6) biological samples/case.

**Figure 3 diagnostics-12-02024-f003:**
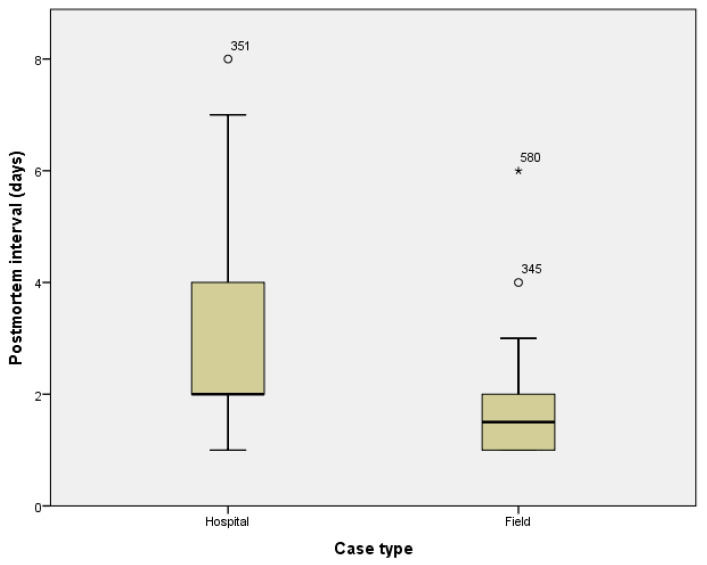
Postmortem interval in days according to case type; postmortem interval for sampling was significantly shorter for field cases compared to hospital cases mean postmortem interval: hospital cases 2.70 ± 1.272 days and field cases 1.83 days ± 1.108, (*p* < 0.001). ‘o, *’ high extreme values.

**Figure 4 diagnostics-12-02024-f004:**
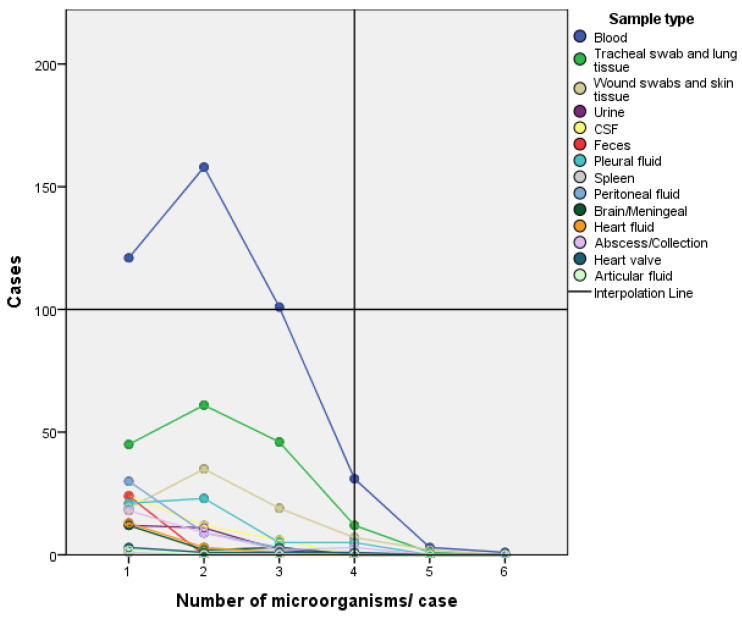
Bacteriology results—positive cultures—number of microorganisms per case depending on sample type: blood cultures were positive for one to six microorganisms, tracheal swab and lung tissue sample cultures and wound swabs and skin tissue cultures were positive for one to five microorganisms; for pleural fluid, spleen and heart valve tissue sample cultures and collection cultures’ positive results found one to four microorganisms/case; peritoneal fluid and pericardial fluid samples’ positive cultures revealed one to three microorganisms/case; urine cultures identified one or two microorganisms; articular fluid, cerebrospinal fluid cultures (CSF), meningeal and brain tissue samples’ positive cultures all identified one microorganism.

**Figure 5 diagnostics-12-02024-f005:**
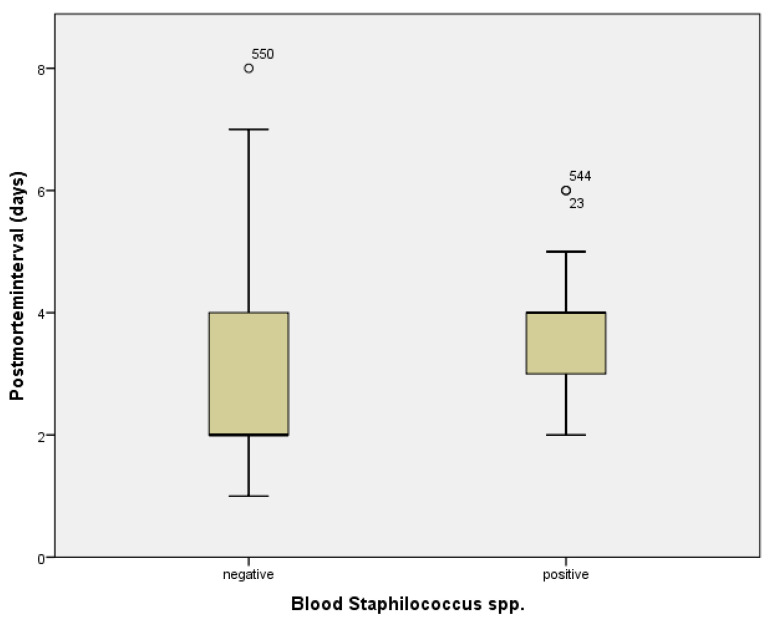
Postmortem intervals in days and Staphylococcus species; ‘o’ high extreme values.

**Figure 6 diagnostics-12-02024-f006:**
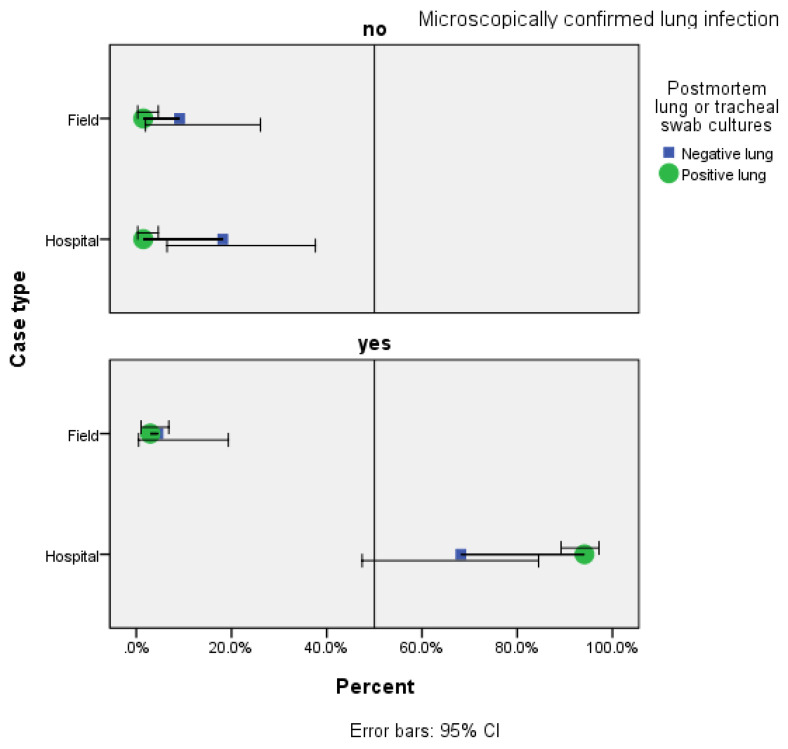
Graphic representations of lung culture results associated with microscopically confirmed lung infection in stratification analysis, with subgroups by case type.

**Table 1 diagnostics-12-02024-t001:** Collected data from archives.

Source	Data
Request for analysis	
	Autopsy date
	Sample type
	Hospital/field case
	Age
	Gender
Medical records	
	Diagnosis of infection
	Hospital bacteriology result
	Hospital
Autopsy report	
	Date of death
	Manner of death
	Cause of death—infection yes/no
	Histopathology result
Bacteriology Results	
	Positive/negative culture
	Microorganism identified

**Table 2 diagnostics-12-02024-t002:** Descriptive statistics of data retrieved from autopsy and hospital records.

		No	%
Infection listed in cause of death	Yes	454	83.2%
	No	92	16.8%
Infection listed in hospital diagnosis	Yes	346	66.3%
	No	176	33.7%
Hospital bacteriology results	No	288	55.2%
	Yes/positive	213	40.8%
	Yes/negative	21	4.0%

**Table 3 diagnostics-12-02024-t003:** The frequency of each sample type in our study population. In the majority of cases, blood cultures were requested—457 cases (72.54%). Second place was held by tracheal swabs—168 cases (26.67%). Peritoneal, pleural and cerebral spinal fluid samples were collected in one out of ten cases (11.90%, 11.11% and 10.63%). Wound swabs, faeces, abscess/fluid collection, pericardial fluid, urine, meningeal, skin, lung and muscle tissue samples were retrieved in fewer than 10% of cases.

Sample Total: 1166	Case Total: 630	
	No %	
Blood	457	72.54%
Tracheal swab	168	26.67%
Peritoneal fluid	75	11.90%
Pleural fluid	70	11.11%
Cerebrospinal fluid (CSF)	67	10.63%
Wound swab	59	9.37%
Faeces	52	8.25%
Abscess/Collection	44	6.98%
Pericardial fluid	39	6.19%
Urine	36	5.71%
Meningeal/ brain tissue sample	24	3.81%
Skin tissue sample	23	3.65%
Lung tissue sample	22	3.49%
Muscle tissue sample	15	2.38%
Spleen tissue sample	7	1.11%
Heart valve tissue sample	5	0.79%
Synovial fluid	3	0.48%

**Table 4 diagnostics-12-02024-t004:** The frequency for bacterial families identified in our culture results.

Bacterial Taxonomy	Positive Cultures											
	Blood		Skin/ Wound		Lung/ Tracheal		Pleural Fluid		Peritoneal Fluid		Abscess	
	No	%	No	%	No	%	No	%	No	%	No	%
*Enterobacteriaceae*	275	60.2	25	30.49	110	66.67	41	58.57	26	34.67	19	42.22
*Enterococcaceae*	185	40.5	46	56.1	40	24.24	25	35.71	25	33.33	8	17.78
*Staphylococcaceae*	77	16.8	20	24.39	24	14.55	-	-	6	8	5	11.11
*Streptococcaceae*	27	5.9	3	3.66	4	2.42	1	1.43	3	4	5	11.11
*Pseudomonas and Acinetobacter baumannii*	148	32.4	51	62.2	105	63.64	17	24.29	17	22.67	9	20

## Data Availability

Not applicable.

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
