# Peer review of "Postmortem Bacteriology in Forensic Autopsies—A Single Center Retrospective Study in Romania"

_diagnostics, 2022, doi:10.3390/diagnostics12082024_

Round 1

Reviewer 1 Report

Authors accomplished all suggested  revisions and  therefore modified consistently their paper report. In the present for this research is worthy of publication.  

Author Response

Thank you for the comments and suggestions

Reviewer 2 Report

Being a resubmission, my comments will be short.

While I appreciate the change in bacterial - since microbiolgy is too wide - I cannot see a general improvement of the paper.

The authors do not discuss the enormous issue, which is the fact that almost all deaths are to be related to bacteria infections. I suggest it is a huge bias authors again failed to explain. 

Moreover, while I appreciate the subdivision of the sample in groups per year, I am a little buffled from the results. May I receive the database so to perform a second analysis?

Author Response

Thank you for the comments and suggestions:

Being a retrospective study we had no control over the autopsy cases for which bacteriological examination was requested by the case legal-medicine specialist and as we stated this is not part of the standard autopsy protocol in our center and it is an additional examination that is done based on the clinical judgment and in accordance with the findings in each case, on a case-by-case basis. From the records the vast majority of the cases for which this examination was requested were hospital cases, severely wounded/burned patients that received medical care but with an unfavorable outcome. We aimed to assess if the results of the bacteriological examinations were considered relevant and included in causes of death (final, intermediate or initial) and the correlation with histological results or if they were considered contamination of any kind and not included in the final conclusions of the autopsy report.

We did not have at our disposal the reasons of the case doctors to solicit the examinations, as it is done on a case-by-case-basis, one can assume it was to confirm/identify an etiological agent or to assess a treatment result. But the aim of our study was not to profile immune-compromised patients as this has been a subject of extensive research in clinical fields and resulted in severity and prognostic scales. Regarding the types of medical units included in our analysis, most cases came from emergency hospitals - these admit a large number of patients with life-threatening conditions

First we would like to send the SPSS file(modif) with the results for the age groups and the database with the comment that it was constructed as part of a PhD thesis in the Romanian language, thus we are attaching it half in Romanian, and half in English (the raw Excel data is in Romanian), please reply if you have any issues understanding. We are sending the files as a link for transfer as the are too large for this section> https://we.tl/t-DdRvco1tnk 

                The assessment of HCAIs was performed for another paper, only on the hospital cases with full data available from which we attach a paragraph

“…77.7% of our study population consisted of critically ill or severely injured patients – with patient deaths occurring in the ICU after a median stay of 10 days. For analysis regarding HCAI, we divided our study population into groups by length of hospital and ICU stay, according to case definitions of previously mentioned infections. Using this threshold (3 days), we identified similar percentages of cases in groups independent of the criterion used. […].., our analysis aimed to exclude these contaminants by focusing on ESKAPE pathogens through selecting sample type results associated with histological infection (over 80% of cases).  Results showed statistically significant associations for ESKAPE with prolonged hospital and ICU stay independent of postmortem interval, thus showing the positive predicting value of the results. Moreover, AMR isolates of the previously mentioned species were associated with prolonged hospital and/or ICU stay.”

Round 2

Reviewer 2 Report

I suggest adding the answer the Authors provided in the discussion section and to modify accordingly both the methods and the conclusion. 

Author Response

Thank you for the comments and suggestions,

We modified methods, discussions, and conclusions thus we attach the revised manuscript as a file.

The assessment of HCAIs was performed for another paper that is awaiting publication.

This manuscript is a resubmission of an earlier submission. The following is a list of the peer review reports and author responses from that submission.

Round 1

Reviewer 1 Report

POSTMORTEM MICROBIOLOGY IN FORENSIC AUTOPSIES – A SINGLE CENTER RETROSPECTIVE STUDY IN ROMANIA. The study was a retrospective,  single-center study, evaluating postmortem microbiology (PMM) on  630 forensic autopsies, 245 female (38.9%) and 385 male (61.1%), age range 0 and 94 years, median age of 52 years.  Deaths occurred in hospital for 594 cases (94.3%) and out-of-hospital for 36 cases (5.7% - field case).  Blood cultures were requested in the majority of cases, followed by tracheal swabs and lung tissue.  In-hospital and out of hospital deaths did not differ significantly regarding the number of microorganisms identified in a positive blood culture. Postmortem microbiology cultures of the respiratory 26 tract showed a statistically significant association to microscopically confirmed lung infections. 

Authors in their conclusions  affirmed  that "Postmortem sampling for microbiology testing in our center in Bucharest is heterogeneous with a high variation of patterns": please specify and detail this sentence.

A positive blood culture result for Staphylococcus species  without the identification of a specific microorganism is more likely due to postmortem contamination:  please specify whether is related to review literature or by personal experience by research. 

Contamination during sampling procedures and postmortem translocation were investigated systematically in one hundred medico-legal autopsies of out-of-hospital 61 deaths: please clarify the meaning of "translocation". 

Please clarify the meaning of the sentence "Without case type stratification", the overall probability of obtaining a positive microbiology result is 12.37 (95% CI: 3.15-48.57) times greater in microscopically confirmed lung infection. 

Please correct  and cite appropriate the reference number 14 as shown  below: 

14. Argo, A., Ventura Spagnolo, E., Zerbo, S., Mondello, C., D’Aleo, F., Conte, M., Stassi, C., & Baldino, G. (2019). Forensic microbiology: A case series analysisEuroMediterranean Biomedical Journal14, 117-121.

Reviewer 2 Report

I would like to thank the Editor for giving me the opportunity of reviewing the article titled Postmortem microbiology in forensic autopsies – a single center retrospective study in Romania.

I would like to outline my review following the structure given by the Authors.

Title

First of all, I think the title itself is quite ambiguous, since the Introduction section is focused on bacteria, which are only one family of the context of microbiology.

Introduction

The background is poor, and the aim not so well emphasised. A further literature examination is necessary.  

Moreover, paragraph at lines 41-47 should be put after the following paragraph (lines 48-55), in order to sound logical.

Line 69: is that a single quotation mark before was suspected? If so, where does the quote end?

Methods

Lines 144-123: the Authors describe the routinary procedure at their Center, however, can they provide the protocol they used? Can they ensure that the same protocol was performed in the considered period, by the same professionals?

Lines 129-133: there is a mixture of methods and results.

Results

Lines 148-149: the study population is necessarily extremely wide as far age is considered. Since the youngest and the eldest are the subjects most commonly affected by bacterial infections, Authors should consider a description for age categories.

Lines 156-157: the sum of the manner of death % is not equal to 100%. Please explain.

Lines 236-258: I think that counting the number of pathogens in the collected sample is useless, because (a) many bacteria can spread and cause sepsis, and (b) many bacteria have a typical localization.

Figures 1, 2, 3 and 4: shouldn’t the description be under the images and not over them?

Discussion

Lines 354-358: I think it is pretty obvious that hospital autopsy have more patterns than the so-called field cases, because – for instance – of healthcare acquired infections. Please expand or remove.

General: In the results section, the Authors state that microbiological analysis was requested up to 5.59% per year (lines 146-147); yet, infection was mentioned in the death cascade in ~80% of cases. This contradiction must be discussed. Moreover, since infection caused, directly or indirectly, almost all deaths, the Author should discuss this issue, since it may be a huge bias.

Round 2

Reviewer 2 Report

.